# The Utility of Endoscopic-Ultrasonography-Guided Tissue Acquisition for Solid Pancreatic Lesions

**DOI:** 10.3390/diagnostics12030753

**Published:** 2022-03-19

**Authors:** Hiroki Tanaka, Shimpei Matsusaki

**Affiliations:** Department of Gastroenterology, Suzuka General Hospital, Suzuka 513-8630, Japan; mossmossgreen2000@yahoo.co.jp

**Keywords:** endoscopic ultrasonography, EUS-guided tissue acquisition, EUS-guided fine-needle aspiration, EUS-guided fine-needle biopsy, solid pancreatic lesions, pancreatic ductal adenocarcinoma, pancreatic neuroendocrine neoplasms

## Abstract

Endoscopic-ultrasonography-guided tissue acquisition (EUS-TA) has been widely performed for the definitive diagnosis of solid pancreatic lesions (SPLs). As the puncture needles, puncture techniques, and sample processing methods have improved, EUS-TA has shown higher diagnostic yields and safety. Recently, several therapeutic target genomic biomarkers have been clarified in pancreatic ductal carcinoma (PDAC). Although only a small proportion of patients with PDAC can benefit from precision medicine based on gene mutations at present, precision medicine will also be further developed for SPLs as more therapeutic target genomic biomarkers are identified. Advances in next-generation sequencing (NGS) techniques enable the examination of multiple genetic mutations in limited tissue samples. EUS-TA is also useful for NGS and will play a more important role in determining treatment strategies. In this review, we describe the utility of EUS-TA for SPLs.

## 1. Introduction

Endoscopic-ultrasonography-guided tissue acquisition (EUS-TA), including EUS-guided fine-needle aspiration (EUS-FNA) and fine-needle biopsy (FNB), has been widely performed to pathologically diagnose solid pancreatic lesions (SPLs). The tissue acquisition, diagnostic accuracy, and safety of EUS-TA for SPLs are superior to those of other procedures such as transpapillary tissue sampling, and EUS-TA is the standard tissue sampling method for SPLs. Furthermore, many efforts are being made to improve the diagnostic accuracy of EUS-TA, such as the selection of better needles, puncture methods, and sample processing methods. Recently, advances in next-generation sequencing (NGS) techniques have enabled the examination of multiple genetic mutations in limited tissue samples, as well as precision medicine based on gene mutations. EUS-TA is also useful for NGS, and is now important not only for pathological diagnosis, but also for treatment decisions based on gene mutations. In this review, we describe the utility of EUS-TA for SPLs, mainly with reference to the literature of the last five years.

## 2. Diagnostic Ability of EUS-TA for Each Pancreatic Solid Lesion

Table 1 shows the histological characteristics of SPLs.

### 2.1. Pancreatic Cancer

Pancreatic ductal adenocarcinoma (PDAC) has a poor prognosis, with patients with an unresectable disease accounting for approximately 80% of all patients with PDAC, and most patients undergo EUS-TA for their diagnosis. Many studies have investigated the diagnostic ability of EUS-TA for PDAC. A meta-analysis of 33 studies published between 1997 and 2009 has reported that the pooled sensitivity and specificity of EUS-FNA (22- or 25-gauge [G]) in diagnosing PDAC were 85% (95% CI: 84–86%) and 98% (95% CI: 97–99%), respectively [1]. Furthermore, a recent meta-analysis of 11 studies published between 2012 and 2018 showed that the pooled sensitivities of EUS-FNA (22G) and EUS-FNB (22G) for the diagnosis of PDAC were 90.4% (95% CI: 86.3–94.5%) and 93.1% (95% CI: 87.9–98.4%), respectively, while their pooled specificities were both 100% [2]. As mentioned above, the diagnostic ability of EUS-TA for PDAC may have improved recently.

EUS-TA is also performed for rare subtypes of pancreatic cancer, including acinar cell and anaplastic carcinoma (Figure 1). The usefulness of EUS-TA for these entities remains unknown due to their rarity; however, some case reports and case series have reported that EUS-TA is useful in diagnosing these subtypes [3,4,5].

### 2.2. Pancreatic Neuroendocrine Neoplasms (PanNENs)

PanNENs are rare pancreatic tumors accounting for 1–2% of all pancreatic neoplasms. EUS-TA is useful for diagnosing PanNENs, and its sensitivity and specificity are reported to be 84.5–98.9% and 99.4–100%, respectively [6,7]. In the 2017 World Health Organization (WHO) classification, PanNENs were categorized as neuroendocrine tumor (NET)-G1, NET-G2, NET-G3, or neuroendocrine carcinoma (NEC) based on their histological features, Ki-67 labeling index (LI), and mitotic count, which is important for predicting prognosis and determining treatment strategies [8]. However, PanNENs show heterogeneity in terms of their Ki-67 LI and mitotic count. EUS-FNA specimens are often underestimated in their Ki-67 LI compared to surgical specimens, and the concordance rates of the WHO grading classification between EUS-TA specimens and resected specimens were from 61–89.4% [6,9,10,11,12,13,14]. Hasegawa et al. showed that a grading classification discrepancy occurred between EUS-TA and the resected specimens when the number of evaluable tumor cells obtained by EUS-TA was small, and the concordance rate was high at 90% (18/20) when ≥2000 tumor cells obtained by EUS-TA were evaluable [15]. To determine the grading of PanNENs, the WHO recommends assessing ≥ 500 tumor cells from hot spots, and the European Neuroendocrine Tumor Society (ENETS) recommends assessing ≥ 2000 tumor cells [8,16]. Therefore, it is important to obtain a sufficient amount of the specimen using EUS-TA for accurate evaluation of Ki-67 LI.

### 2.3. Solid Pseudopapillary Neoplasms (SPNs)

SPNs are a rare pancreatic primary tumor with low malignant potential, but they can occasionally develop distant metastases. The sensitivity of EUS-TA in the diagnosis of SPNs is from 80.8–82.6% [17,18]. Some pathological features of SPNs, such as their small, round nuclei and numerous microvessels, are similar to those of PanNETs, and SPNs are often positive for endocrine markers, including synaptophysin and chromogranin A. Therefore, it is sometimes difficult to differentiate between SPNs and PanNETs by cytomorphology alone, and immunohistochemistry (IHC) is also important [17]. β-catenin nuclear labeling is also useful in differentiating SPNs from PanNETs [19]. Additionally, Foo et al. reported that immunostaining with SOX-11 and TFE3 using EUS-TA specimens was useful in differentiating SPNs from PanNETs [20].

### 2.4. Metastatic Tumors to the Pancreas

The primary sites of metastatic tumors to the pancreas include renal cell carcinoma, lung cancer, malignant melanoma, breast cancer, colon cancer, bladder cancer, and others [21,22,23]. Pathological diagnosis is important because metastatic tumors to the pancreas are sometimes difficult to differentiate from primary pancreatic tumors by imaging. Since most metastatic tumors to the pancreas are not indicated for surgical resection, EUS-TA is often performed. The sensitivity and specificity of EUS-TA for diagnosing metastatic tumors to the pancreas are 84.9–95.9% and 100%, respectively, which are comparable to the results of EUS-TA for primary pancreatic tumors [21,22,23]. A combination of cytomorphology and IHC based on the characteristics of the primary tumor is useful for identifying the primary tumor site (Figure 2 and Figure 3) [21,22,23].

### 2.5. Autoimmune Pancreatitis (Mass-Forming Pancreatitis)

Autoimmune pancreatitis (AIP) is a distinct form of pancreatitis that is classified into two types, diffuse and segmental/focal. In particular, the focal type is considered a type of mass-forming pancreatitis and is sometimes difficult to differentiate from PDAC due to its abundant fibrosis. AIP shows characteristic pathological findings called lymphoplasmacytic sclerosing pancreatitis including lymphocyte infiltration, IgG4-positive plasma cells, storiform fibrosis, and obliterative phlebitis. Core tissue samples are required for evaluating these pathological findings, and EUS-FNA is not suitable for diagnosis, as its sensitivity is only 54–63% [24,25,26]. EUS-FNB with Franseen- or Fork-tip-type needles is particularly useful for the diagnosis of AIP, with a sensitivity of 78–93% [27,28,29]. Furthermore, the presence of extrapancreatic lesions is also helpful in the diagnosis of AIP.

### 2.6. Primary Pancreatic Lymphoma (PPL)

PPL is an extremely rare pancreatic tumor accounting for 0.5% of all SPLs [30], and subtypes such as diffuse large cell lymphoma, follicular lymphoma, mantle cell lymphoma, and others have been reported. PPL is usually depicted as a heterogeneous hypoechoic mass on US or EUS, and is sometimes difficult to differentiate from PDAC. Furthermore, diagnosis of only the lymphoma itself is not sufficient, as diagnosis of its subtypes is also important for treatment-strategy decisions and prognosis predictions. Flow cytometry (FCM) is known to play an important role in diagnosing lymphoma and its subtypes. Although some case series have reported the utility of EUS-TA for PPL, the sensitivity of EUS-FNA cytology alone was only 28–30.6% [30,31,32]. On the other hand, FCM analysis in combination with EUS-FNA cytology improved the sensitivity of the PPL diagnosis (75–100%) compared to cytology alone. Additional passes and sample processing methods for FCM analysis should be considered if onsite evaluations of EUS-TA specimens suggest the possibility of PPL.

**Table 1 diagnostics-12-00753-t001:** Histological characteristics of solid pancreatic lesions.

	Cytology/Histology	Immunohistochemistry (IHC)	Genetic Abnormalities
PDAC	Desmoplasia	KRAS, p53, Dpc4, p16	*KRAS*, *TP53* *SMAD4*, *CDKN2A*
Acinar cell carcinoma	Acinar structure Glandular structure Cribriform pattern	Trypsin, BCL10	*SMAD4*, *JAK1*, *BRAF* *BRCA2*, *FAT*, *CTNNB1*, *APC*
Anaplastic carcinoma	Pleomorphic type Spindle type Osteoclast-like giant cells	Keratin, CK7/20, Vimentin	*KRAS*, *TP53*
PanNET	Well-differentiated Mitotic count	Ki-67 labeling index SSTR2A, DAXX, ATRX	*DAXX*, *ATRX*
PanNEC	Poorly differentiated	Ki-67 labeling index Rb, p53	*KRAS*, *RB1*, *TP53*
SPN	Differential diagnosis: PanNET	β-catenin nuclear labeling SOX-11, TFE3	*CTNNB1*
Metastatic tumors to the pancreas	Similar to the primary tumor	Depending on characteristics of the primary tumor	
AIP	Lymphocyte infiltration Storiform fibrosis Obliterative phlebitis (Victoria blue staining)	IgG4-positive plasma cells	
Pancreatic lymphoma	Low sensitivity	CD20, CD79a, CD5, CD10, CD3 Cyclin D1, bcl-2, TdT	Depending on the subtype

PDAC, Pancreatic ductal adenocarcinoma; PanNET, Pancreatic neuroendocrine tumor; PanNEC, Pancreatic neuroendocrine carcinoma; Solid pseudopapillary neoplasm, SPN; AIP, Autoimmune pancreatitis.

## 3. How to Improve the Diagnostic Performance of EUS-TA

Table 2 shows the efforts to improve the diagnostic accuracy of EUS-TA.

### 3.1. Choice of Puncture Needles

EUS-FNA needles (Menghini type) were mainly proposed for cytology initially, and are reportedly useful in diagnosing SPLs (Table 3), with sensitivities and accuracies of 59.6–85% and 63.5–93.3%, respectively [33,34,35,36,37]. Regarding diagnostic abilities based on FNA needle size, Facciorusso et al. conducted a network meta-analysis comparing 22G and 25G FNA needles, which showed that the diagnostic accuracy, sensitivity, specificity, sample adequacy, and histologic core procurement of 22G and 25G FNA needles were similar [38]. Few studies have compared 19G and 22G FNA needles. Song et al. suggested that the 19G needle was superior to the 22G needle in diagnostic accuracy (94.5% vs. 78.9%, *p* = 0.015) by per-protocol analysis, excluding cases of technical failure [39]. The 19G needle is thought to have a lower success rate than the 22G needle, and a 19G nitinol needle has recently been developed to address this problem. However, EUS-FNA with a 19G nitinol needle is sometimes difficult in the transduodenal approach (technical success rate with 19G: 86.4% vs. with 22G: 100%, *p* = 0.003) [40]. Therefore, tumor location and puncture route should be considered when selecting puncture needles.

EUS-FNB needles have been developed to improve specimen acquisition and diagnostic abilities. Although there is heterogeneity in the results among studies, EUS-FNB is generally superior to EUS-FNA in terms of diagnostic accuracy, specimen adequacy, and number of needle passes, and is equivalent in adverse events [41]. EUS-FNB needles are classified into Franseen, Fork-tip, Forward-bevel, and Reverse-bevel types according to the shape of the needle tip. Several studies compared the usefulness of EUS-FNB needles. Karsenti et al. showed that the Franseen type was superior to the Forward-bevel type in terms of diagnostic accuracy (87% vs. 67%), diagnostic adequacy (100% vs. 82%), length of the tissue core (11.4 mm vs. 5.4 mm), and surface area of the tissue core (3.5 mm^2^ vs. 1.8 mm^2^) [42]. Furthermore, Crino et al. showed that the Fork-tip type was equivalent to the Reverse-bevel type in diagnostic accuracy (92.7% vs. 91.7%), superior in core specimen acquisition (54.2% vs. 6.3%), and had a lower blood contamination (<25% of the slide) (79.2% vs. 38.5%) [43]. A meta-analysis also showed that the Franseen and Fork-tip types were superior to the Reverse-bevel type in terms of diagnostic accuracy [41]. In studies comparing the Franseen and Fork-tip types, the diagnostic accuracy (92.3% vs. 94.4%), diagnostic adequacy (94.9–96% vs. 92–97.2%), surface tissue area (6.1 mm^2^ vs. 8.2 mm^2^), and surface tumor area (0.9 mm^2^ vs. 1.0 mm^2^) were similar [44,45]. Regarding the relationship between utility and FNB needle size, Tomoda et al. compared 22G and 25G Franseen-type needles, and the 25G needle was equivalent to the 22G needle in its adequate tissue acquisition rate (70.5% vs. 78.4%), sensitivity (84.5% vs. 86.9%), and diagnostic accuracy (86.4% vs. 89.8%) [46]. Because of the differences in the study designs and puncture methods in each of the abovementioned studies, further studies are needed to determine the appropriate puncture needle.

### 3.2. Puncture Methods

#### 3.2.1. Door-Knocking Method, Fanning Technique, and Suction Techniques

Puncture techniques, including the door-knocking method and the fanning technique, and suction techniques, including the high-negative-pressure method, slow-pull method, and wet-suction method, have been developed to improve the technical success, sample adequacy, and diagnostic ability of EUS-TA. The door-knocking method is a quick needle advancement technique for target lesions and is useful in high-cellularity tissue acquisition by transgastric puncturing [47]. However, the door-knocking method in the transduodenal approach or for small lesions was inferior to the conventional method in terms of adequate tissue acquisition and diagnostic accuracy, and indications for the door-knocking method should be determined based on the puncture site and tumor diameter. The fanning technique was used to obtain specimens from different parts of the tumor by gradually changing the angle of the puncture needle within the tumor. Bang et al. suggested that the fanning method has the advantage of requiring fewer punctures for diagnosis than conventional methods (number of passes, mean 1.2 vs. 1.7, *p* = 0.02) [48].

There have been two meta-analyses of suction techniques. Ramai et al. compared wet and dry suction, and wet suction has shown advantages in terms of specimen adequacy (pooled OR: 3.18, 95% CI: 1.82–5.54) [49]. Wang et al. showed that the slow-pull technique was better than the standard method in terms of core tissue acquisition (pooled OR: 1.91, 95% CI: 1.11–3.26), blood contamination (pooled OR: 1.93, 95% CI: 1.29–2.87), and diagnostic accuracy (pooled OR: 1.60, 95% CI: 1.14–2.26) [50]. Various innovations to the EUS-TA puncture method have been made as described above, but there is no consensus on the appropriate puncture method in EUS-FNA so far because each method has not yet been sufficiently compared.

#### 3.2.2. Number of Passes

Additional punctures reportedly improve the sensitivity when initial EUS-TA is inconclusive. Several studies have investigated the optimal number of passes for the diagnosis of SPLs, with t improvements in the sensitivity of EUS-TA reaching a plateau after the third or fourth puncture, regardless of tumor localization or size [51,52,53,54]. Therefore, other examinations should be considered if the diagnosis is inconclusive even after the third or fourth puncture. On the other hand, recent advances in puncture needles (especially the Franseen and Fork-tip types) have made it possible to make a diagnosis with fewer punctures, and the sensitivity is reported to reach a plateau after the second puncture [24,27,35].

#### 3.2.3. EUS-TA with Contrast Enhanced EUS (CE-EUS)

SPLs, including PDAC and PanNEC-G3, often exhibit necrosis and abundant fibrous stroma, which may result in false-negative EUS-FNA results. CE-EUS is useful for distinguishing between viable and necrotic areas of SPLs, and EUS-FNA with CE-EUS (CE-EUS-FNA) has been attempted to improve the diagnostic ability of SPLs by avoiding non-enhancing areas within the lesions. Several studies have assessed the utility of CE-EUS-FNA, and the accuracy, sensitivity, and specificity of EUS-FNA with CE-EUS are comparable or superior to those of EUS-FNA alone [53,55,56,57,58]. Furthermore, a recent meta-analysis revealed that the accuracy (88.8% vs. 83.6%), sensitivity (84.6% vs. 75.3%), and sample adequacy (95.1% vs. 89.4%) of CE-EUS-FNA are superior to those of EUS-FNA alone [59]. Itonaga et al. classified contrast enhancement patterns into three subgroups (heterogeneous (tumor tissues], homogeneous (inflammatory tissues], and non-enhancing (necrotic tissues] areas), and reported that CE-EUS-FNA that avoided the homogeneous and non-enhancing areas improved its sensitivity and sample adequacy [58].

#### 3.2.4. EUS-Elastography

EUS-elastography is also performed as an ancillary imaging study and is classified into two categories: strain elastography, which measures the strain caused by pressure in response to compression, and shear wave elastography, which measures the velocity of shear wave propagation. EUS-elastography has been shown to be useful in differentiating malignant and benign pancreatic lesions based on their difference in tissue elasticity. Furthermore, EUS-elastography-guided EUS-FNA can improve its diagnostic ability by targeting stiffer areas within the same mass lesion. Facciorusso et al. evaluated the utility of EUS-elastography-guided EUS-FNA for SPLs, and its diagnostic accuracy, sensitivity, and specificity were favorable (94.4%, 93.4%, and 100%, respectively) [60]. This study was a retrospective single-arm study, and further studies are needed to determine the additive effect of EUS-elastography on EUS-FNA.

### 3.3. Sample Processing Methods

#### 3.3.1. Rapid On-Site Evaluation (ROSE)

ROSE is an immediate cytology using Diff-Quik staining or ultrafast Papanicolaou staining and has been performed to improve the sample adequacy and diagnostic ability of EUS-TA. ROSE can make an immediate cytological assessment of EUS-TA specimens, and the number of FNA passes can be determined based on the presence of tumor cells and the amount of specimen evaluated by ROSE. A recent retrospective study compared the sensitivity of EUS-FNA/FNB with and without ROSE in diagnosing SPLs [61]. EUS-FNA with ROSE has better sensitivity than EUS-FNA alone (91.96% vs. 70.83%, *p* < 0.001). The sensitivity of EUS-FNB without ROSE was also superior to that of EUS-FNA without ROSE (87.44% vs. 70.83%, *p* < 0.001); however, there was no significant difference in the sensitivity between EUS-FNA with ROSE and EUS-FNB without ROSE (91.96% vs. 80.72%, *p* = 0.193). Furthermore, Khan et al. conducted a meta-analysis and showed that EUS-FNB was associated with relatively better diagnostic adequacy than EUS-FNA, but no significant difference between FNA + ROSE and EUS-FNB was observed [62]. ROSE can also reduce the number of EUS-FNA needle passes and is expected to reduce the complication rate. Although ROSE is considered useful, as described above, there was heterogeneity in the results among studies, and no significant difference in diagnostic ability between EUS-FNA with and without ROSE has been found in some studies [63,64]. In particular, EUS-FNB with the new-generation needles already has a high diagnostic accuracy, and the additional effect of ROSE on diagnostic ability may be small [64]. ROSE is a time- and human-resource-consuming examination, and further investigation is needed to determine the cases in which ROSE is more effective.

#### 3.3.2. Macroscopic On-Site Evaluation (MOSE)

Despite the utility of EUS-FNA with ROSE, ROSE has not been uniformly performed in all centers because of limited human pathological resources. Therefore, MOSE is as an alternative method when ROSE is unavailable. EUS-TA specimens mainly contain white core specimens, red core specimens, and blood, and the visible white cores of the EUS-FNB sample contain histological cores more frequently than red cores [65]. In MOSE, the specimen is macroscopically evaluated for the presence and length of visible whitish cores. The diagnostic ability was improved depending on the visible whitish core length evaluated by MOSE, and a sufficient length of visible whitish cores reflected a high diagnostic ability [65,66]. However, the cut-off values for the optimal length of macroscopic visible white cores vary among studies, such as 4 mm and 10 mm [65,66], and remain inconclusive. In any case, MOSE may replace ROSE in cases where ROSE is unavailable. On the other hand, no studies have directly compared ROSE and MOSE in EUS-FNB. At present, either ROSE or MOSE should be considered in accordance with the standards of each hospital.

#### 3.3.3. Liquid-Based Cytology (LBC)

In addition to efforts to obtain adequate specimens, efforts have also been made to diagnose SPLs with limited specimens, including LBC. The advantages of LBC include the ability to collect a larger number of tumor cells, concentrate the tumor cells with a thin layer, and reduce blood contamination and cell crowding. The accuracy, sensitivity, and specificity of LBC in EUS-FNA are comparable or superior to those of the conventional smear method (CS) [67,68,69]. Zhou et al. showed that the combination of LBC with CS in EUS-FNA improved the diagnostic ability for SPLs (the sensitivity of CS, LBC, and CS + LBC was 55.1%, 71.4%, and 83.9%, respectively, and the accuracy of CS, LBC, and CS + LBC was 61.6%, 76.1%, and 86.5%.) [68]. Sekita-Hatakeyama et al. examined the utility of LBC using residual liquid specimens after separating the solid specimens for cell blocks obtained by EUS-FNA. The combination of *KRAS* mutation analysis using residual LBC specimens with cell blocks reportedly improves the sensitivity and accuracy of PDAC diagnosis from 77.4% and 81.3% to 90.3% and 90.7% compared to using cell blocks alone, respectively [70]. Furthermore, residual LBC specimens were useful for NGS, and the *TP53*, *CDKN2A*, *SMAD4*, and *PIK3CA* mutations, in addition to the *KRAS* mutations, could be assessed [71]. LBC in EUS-FNA is mainly divided into precipitation (SurePath) and filtration methods (ThinPrep, CellPrep). Chandan et al. conducted a meta-analysis that compared precipitation LBC, filtration LBC, and CS in EUS-FNA for SPLs. In terms of diagnostic ability, precipitation LBC was superior to CS (CS vs. precipitation LBC; OR = 0.39, *p* = 0.01), and CS was superior to the filtration LBC method (CS vs. filtration LBC; OR = 1.69, *p* = 0.04). However, the LBC method was determined by each center, and direct comparisons between precipitation LBC and filtration LBC have not been performed. The optimal LBC method for EUS-FNA remains controversial.

#### 3.3.4. IHC, Genetic Analysis

Various gene mutations are associated with PDAC, and *KRAS* (78.9–96%), *TP53* (32.1–78%), *SMAD4* (3.6–31%), and *CDKN2A* (3.6–44%) are representative [72,73,74,75,76,77,78,79,80]. Among these, *KRAS* is known to be the most frequent genetic mutation and is strongly suggestive of PDAC. *KRAS* mutation testing reportedly improves the diagnostic performance for PDAC. In fact, a meta-analysis showed that the combination of the *KRAS* mutation with EUS-FNA increased the pooled sensitivity of PDAC diagnosis from 80.6% (95% CI: 72.1–86.9%) to 88.7% (95% CI: 83.6–92.4%) compared with EUS-FNA alone [81]. Recently, it has been suggested that NGS analysis using EUS-FNA specimens may be useful for the diagnosis of PDAC, especially in cases that are suspicious for PDAC but have an inconclusive cytological diagnosis [78,79].

In PanNENs, PanNETs have relatively frequent immunohistochemical abnormalities such as DAXX (9.1–33%) and ATRX (11.1–36.4%), and frequent genetic mutations such as *DAXX* (11.1–25%) and *ATRX* (10–20.7%) (Table 4) [81,82,83,84,85,86,87,88]. In contrast, immunohistochemical abnormalities such as Rb (41.7–73.7%) and p53 (75.0–94.7%), as well as the *KRAS* (28.6–48.7%), *RB1* (71.4%), and *TP53* (57.1–66.7%) mutations, are observed in PanNEC, and these findings overlap with those for PDAC [82,88,89]. IHC and genetic analysis performed on EUS-TA specimens, in combination with cytology and histology, may be useful in diagnosing PanNENs (Table 4).

## 4. EUS-TA for Precision Medicine

### 4.1. Precision Medicine for PDAC

Several genomic biomarkers are therapeutic targets for PDAC. (Table 5) Homologous recombination deficiency (HRD) is the most common genomic abnormality. BRCA1 and BRCA2 are key proteins involved in homologous recombination (HR), and play important roles in DNA double-strand break repairing. The *BRCA 1* and *BRCA 2* mutations lead to HRD, which prevents DNA double-strand break repairing and induces apoptosis. Hence, PDAC with HRD is reportedly more sensitive to platinum-based chemotherapy, which induces a DNA-damaging effect [90]. In addition to *BRCA 1* and *BRCA 2*, HR-related genes include *PALB2, ATM, ATR*, and others, and a meta-analysis showed that the pooled prevalence of HR-related gene mutations (germline and somatic mutations) in PDAC was: *BRCA* 1, 0.9%; *BRCA 2*, 3.5%*; PALB2* 0.2%*; ATM*, 2.2%; and *ATR*, 0.1% [91]. DNA base excision repairing, as well as HR, is also an important DNA repairing pathway and is accelerated by poly (ADP-ribose) polymerases (PARPs). Due to impaired DNA base excision repairing, pancreatic cancer with HRD is also sensitive to PARP inhibitors. In fact, olaparib, a PARP inhibitor, has shown efficacy for maintenance chemotherapy after a platinum-based regimen with germline *BRCA 1-* or *BRCA 2*-mutated PDAC in a phase 3 study (POLO trial) [92]. However, the efficacy of PARP inhibitors in pancreatic cancer with somatic *BRCA 1/2* mutation or other HR-related gene mutations remains unknown.

Although not specific to PDAC, microsatellite instability-high (MSI-H)/ mismatch repair deficiency (dMMR) and neurotrophin receptor tyrosine kinase (*NTRK*) gene fusions are genetic biomarkers for tumor-agnostic treatments.

DNA mismatch repairing (MMR) is a DNA repairing mechanism essential for maintaining genomic stability by restoring DNA replication errors. Mismatch repairing deficiency leads to the accumulation of DNA replication errors in microsatellites (MSI-H). Tumors with MSI-H/dMMR express a high burden of neoantigens and induce an antitumor immune response via T-cell recognition of tumor-specific neoantigens. However, tumor cells express programmed death-1 (PD-1) ligands (PD-L1/PD-L2) on the surface, inhibit the activation of T cells, and induce the immune tolerance of tumor cells by the binding of PD-L1 to the PD-1 expressed on T cells. Therefore, PD-L1/PD-1 blockade inhibitors are expected to reactivate T cells and enhance antitumor immune responses. In a phase 2 study, pembrolizumab, which is a PD-1 inhibitor, showed antitumor efficacy among patients with MSI-H/dMMR advanced solid tumors, including PDAC, and the CR, PR, and SD were 9.9%, 24.5%, and 18.0%, respectively [93]. However, the ORR of pembrolizumab for MSI-H/dMMR PDAC was relatively low, accounting for 18.2% (4/22). In contrast, Le et al. suggested that pembrolizumab was effective (ORR 62%, 5/8), although only a small number of patients with MSI-H/dMMR PDAC were enrolled [94]. The efficacy of pembrolizumab for MSI-H/dMMR PDAC remains controversial, partly because of the rarity of MSI-H/dMMR PDAC (1–2%) [94,95].

*NTRK* gene fusions, including *NTRK1*, *NTRK2*, and *NTRK3*, are primary oncogenic drivers that promote tumor cell proliferation and survival by activating signaling pathways, including the *MAPK*, *PI3K*, and *PLCγ* pathways. The frequency of *NTRK* gene fusion in PDAC is reportedly ≤ 1% [96]. In an integrated analysis of three phase 1/2 trials (ALKA-372-001, STARTRK-1, and STARTRK-2), entrectinib, which is a *ROS-1*/*TRK* inhibitor, has shown efficacy for *NTRK* fusion-positive solid tumors, with an ORR, DCR, median response duration, and median PFS of 57%, 74%, 10.4 months, and 11.2 months, respectively [97]. Larotrectinib was also effective for *NTRK* fusion-positive solid tumors, with an ORR, DCR, median response duration, and median PFS of 79%, 91%, 35.2 months, and 28.3 months, respectively [98].

Although its efficacy in PDAC remains unknown, sotorasib, which targets the *KRAS* G12C mutation, has been developed and has shown anticancer activity in patients with KRAS-G12C-mutated non-small-cell lung cancers [99]. The *KRAS* G12C mutation is also found in a small fraction of pancreatic cancers, and sotorasib showed efficacy in pancreatic cancer in a phase 1 study [100]. Further investigation of the efficacy and safety of sotorasib in pancreatic cancer is warranted.

### 4.2. NGS Using EUS-FNA/FNB Specimens 

As mentioned above, therapeutic target genomic biomarkers are gradually being identified for PDAC. Therefore, the National Comprehensive Cancer Network (NCCN) guidelines recommend germline and tumor/somatic gene profiling for patients with locally advanced and metastatic pancreatic cancer with indications for anticancer therapy [101]. Most pancreatic cancers are unresectable, and therefore EUS-TA has an increasingly important role not only in pathological diagnosis, but also in the decision to treat PDAC. Recently, NGS has been widely performed for the genetic analysis of SPLs because advances in NGS technology have made it possible to analyze multiple genetic mutations in limited tissue samples at a relatively low cost compared with traditional sequencing methods. Several studies have examined the utility of NGS using EUS-TA specimens, and the success rate, or the adequate tissue rate, of NGS analysis in SPLs was favorable, accounting for 57.4–100% of tissues [72,73,74,75,76,77,79,80,102,103]. (Table 6) The variation in the success rate of NGS analysis may be due to differences in the amount or concentration of DNA extracted and the requirements for NGS (proportion of nuclei derived from tumor cells, and the amount or concentration of input DNA). Regarding the amount of DNA extracted, Park et al. reported that the mean extracted DNA amounts in an NGS success group and an NGS failure group were 540 ng and 142 ng, respectively, and the success rate of NGS analysis was improved from 57.4% (109/190) to 76.2% (109/143) when the amount of extracted DNA was more than 50 ng [76]. Hence, a sufficient amount and concentration of extracted DNA would improve the success rate of NGS analysis. Various preanalytical factors, including tumor cellularity, tumor fractionation, and tumor viability, are associated with DNA quality, and the specimen collection and processing methods are important for this reason. EUS-FNB and needle size are considered predictors of successful NGS in SPLs. Some studies have compared the adequate tissue rate of NGS analysis in SPLs between EUS-FNA and EUS-FNB, and showed that EUS-FNB was suitable for NGS analysis (EUS-FNA: 14–66.9% vs. EUS-FNB: 70.4–90.9%) [75,102,103]. Furthermore, the success rate of NGS analysis in SPLs was better with 19G or 22G than with 25G (OR 2.19, 95% CI: 1.08–4.47) [76]. There are also some important notes regarding the pre-analytical specimen processing methods. Formalin-fixed paraffin-embedded (FFPE) DNA extraction methods may cause DNA fragmentations and chemical modifications. Therefore, the formalin fixation time should be shortened for small specimens such as EUS-FNA specimens. In addition, the quality of nucleic acids deteriorates with time, and the most recently collected specimen is suitable for NGS only if it has been collected several times.

The requirements for NGS are important factors in NGS analysis. Some studies have attempted NGS in all cases with SPLs, whereas others have not attempted NGS if its requirements are not fulfilled. In the latter situation, NGS analysis may not have been performed in cases in which NGS analysis was originally possible. Before NGS analysis, pathologists usually select tumor-rich areas of the tissue and evaluate the tumor fractionation to ensure the adequacy of NGS. The amount of input DNA required for NGS depends on the platform, ranging from approximately 10–300 ng. It is estimated that approximately 2000 tumor cells are needed to obtain 10 ng of DNA [104]. NGS analysis may be performed on a larger number of cases by understanding the amount of DNA required for the platform to be used and estimating the amount of DNA extracted from the specimens.

## 5. Adverse Events

In addition to its high diagnostic yield, EUS-TA is a safe procedure, with a low adverse event rate of about 0.5–2%, which is comparable to EUS-FNA and EUS-FNB [41,105,106,107]. Major adverse events associated with EUS-TA for SPLs include pancreatitis, bleeding, peritonitis, and leakage of pancreatic juice; however, most complications improve with conservative treatments. There are some studies regarding the adverse events of EUS-TA. (Table 7) Kanno et al. reviewed 13,566 EUS-FNA cases and showed that EUS-TA for PanNETs may increase adverse events of pancreatitis compared to EUS-TA for PDACs [105]. Li et al. conducted a meta-analysis regarding EUS-FNB-related adverse events. They showed that EUS-FNB with 22G or 25G needles had a low adverse event rate compared with EUS-FNB with 19G needles, and the lesion size and number of passes did not affect the adverse event rate [107]. On the other hand, another study found that the number of passes and to-and-fro movements (>15) were risk factors for EUS-FNB-related pancreatitis. Therefore, the number of passes should be minimized. Some recent studies have shown that Franseen and Fork-tip needles also had a low adverse event rate comparable to other FNB needles [33,34,36,44]. The most serious adverse event of EUS-TA for PDAC is needle tract seeding (NTS), which involves tumor cell implantation along the needle tract. In transduodenal EUS-TA for pancreatic head cancer, the puncture route is usually excised by a subsequent surgery. However, if the needle tract sites are located in the stomach, they may not be surgically removed. Therefore, most cases of needle tract seeding using EUS-TA are pancreatic body or tail cancers. Although the incidence of NTS following EUS-TA has been considered an extremely rare complication, Yane et al. reported a non-negligible incidence for EUS-FNA-related NTS of 3.4% (6/176) in patients who underwent distal pancreatectomy for PDAC [108]. In contrast, preoperative EUS-FNA reportedly does not affect recurrence and overall survival in pancreatic body or tail cancers [108,109]. Although preoperative EUS-TA for pancreatic body and tail cancers remains controversial, it is necessary to minimize the number of EUS-TA punctures in consideration of the possibility of NTS.

## 6. Conclusions

In summary, our recommendations for EUS-FNA of SPLs based on this review are shown in Figure 4. EUS-TA for SPLs has a favorable diagnostic ability and is a safe procedure with a low complication rate. Precision medicine will be further developed for SPLs as more therapeutic target genomic biomarkers are identified. Therefore, EUS-TA will play a more important role in determining the treatment strategy, and it is necessary to select puncture needles, puncture methods, and specimen processing methods in anticipation of NGS analysis when performing EUS-TA.

## Figures and Tables

**Figure 1 diagnostics-12-00753-f001:**
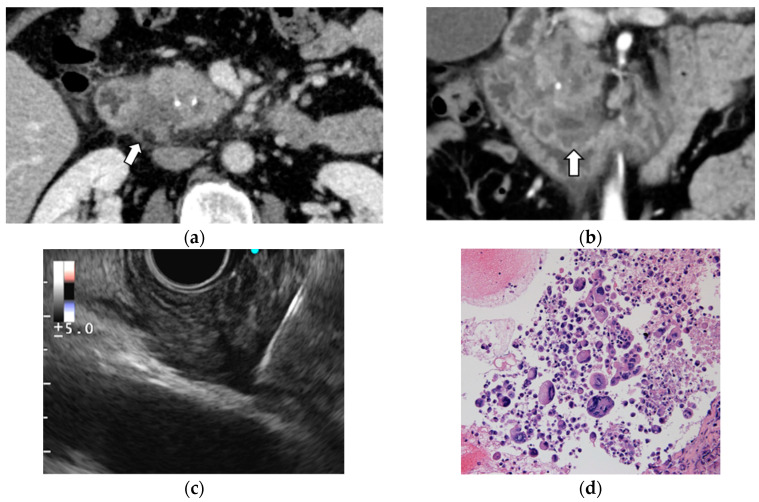
Anaplastic carcinoma of the pancreas. (**a**) Contrast-enhanced computed tomography (CE-CT) demonstrated a poorly circumscribed, low-density mass lesion at the pancreatic head concomitant with chronic pancreatitis (arrow). (**b**) Multiplanar reconstruction (MPR) of CE-CT image. (**c**) EUS-FNA was performed using a Menghini-type 22-gause needle with the fanning technique, and negative pressure was applied with a 20 mL syringe. (**d**) Histology of the specimens by EUS-FNA showed osteoclastic polynuclear giant cells. (hematoxylin and eosin staining, ×400).

**Figure 2 diagnostics-12-00753-f002:**
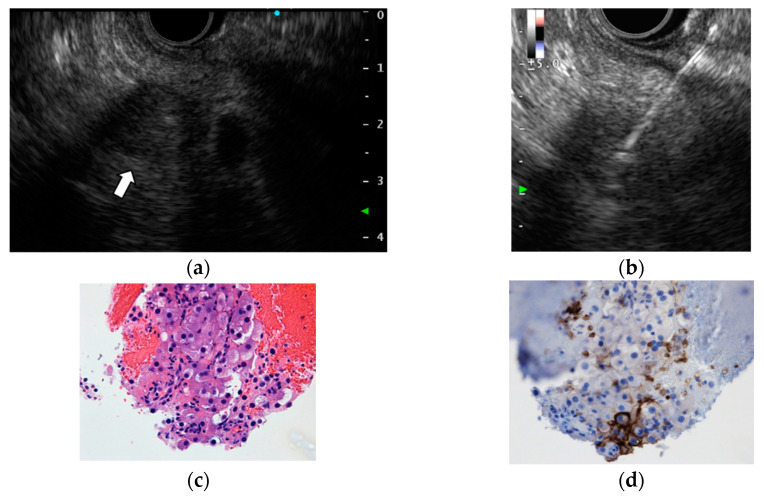
Metastatic renal cell carcinoma to the pancreas. (**a**) Endoscopic ultrasonography (EUS) revealed a well-circumscribed, homogenous low-echoic mass lesion 17 mm in size at the pancreatic tail (arrow). (**b**) EUS-FNA was performed using a Franseen-type 22-gause needle with the fanning technique, and negative pressure was applied with a 20 mL syringe. (**c**–**f**) Hematoxylin and eosin staining showed tumor cells with clear cytoplasm (**c**). In immunohistochemistry, the tumor cells were positive for CD10 (**d**), PAX8 (**e**) and Vimentin (**f**).

**Figure 3 diagnostics-12-00753-f003:**
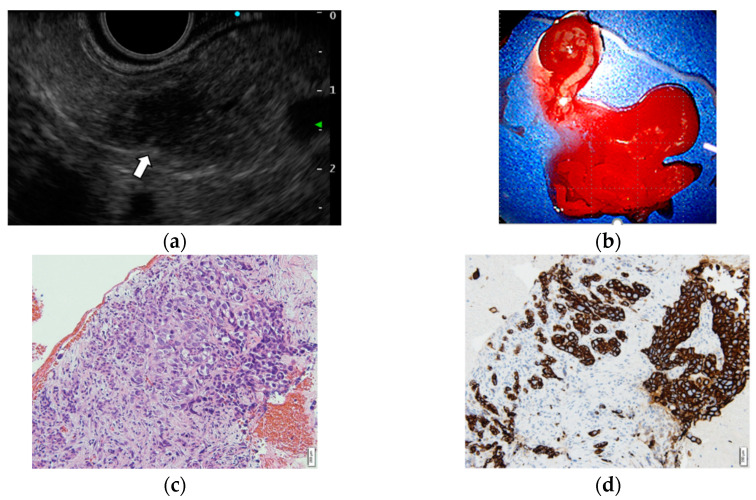
Metastatic bladder carcinoma to the pancreas. (**a**) Endoscopic ultrasonography (EUS) revealed a well-circumscribed, homogenous low-echoic mass lesion 8 mm in size at the pancreatic tail (arrow). (**b**) EUS-FNA was performed using a Franseen-type 22-gause needle, with negative pressure applied with a 20 mL syringe. The specimen included white core tissue, red core tissue, and a liquid component. (**c**) Hematoxylin and eosin staining findings were suspicious for urothelial cell carcinoma. (**d**) The tumor cells were positive for GATA3.

**Figure 4 diagnostics-12-00753-f004:**
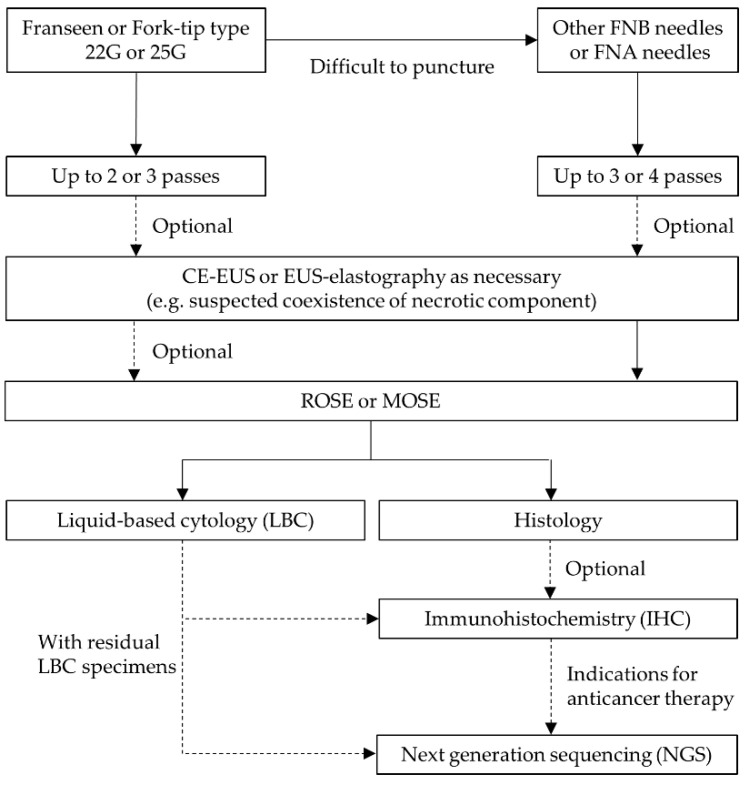
Our recommendations for EUS-FNA of solid pancreatic lesions based on this review.

**Table 2 diagnostics-12-00753-t002:** Efforts to improve diagnostic accuracy in EUS-TA.

		Advantages	Disadvantages
Selection of puncture needles	FNA needles	Relatively easy to puncture	Sometimes insufficient specimen
	FNB needles	Favorable diagnostic ability and tissue acquisition Reduction in the number of punctures	Rarelydifficult to puncture
Puncture methods	Puncture Door-knocking method Fanning technique Suction High-negative-pressure method Slow-pull method Wet-suction method	No consensus on the appropriate puncture method	
Number of punctures		Additional punctures improve the sensitivity	The sensitivity reached a plateau after the 3rd or 4th puncture (Franseen and Fork-tip type: 2nd puncture)
Ancillary imaging studies	CE-EUS EUS-elastography	Improvement of sensitivity and sample adequacy	>Dependent on endosonographer’s experience
On-site evaluation	ROSE MOSE	Reduction in the number of punctures	Time- and human-resource- consuming examination
Sample processing method	LBC	Collection of a larger number of tumor cells with limited specimens Standardization of the sample processing method	Time- and cost- consuming examination
IHC Genetic analysis		Particularly useful in cases of inconclusive cytological diagnosis	A sufficient specimen required

CE-EUS, Contrast enhanced EUS; ROSE, Rapid on-site evaluation; MOSE, Macroscopic on-site evaluation; LBC, Liquid-based cytology; IHC, Immunohistochemistry.

**Table 3 diagnostics-12-00753-t003:** List of puncture needles for EUS-TA.

Needle Type	Main Products	Needle Size	Launched Year in Japan	Diagnostic Accuracy Tissue Acquisition	Ease of Puncture
EUS-FNA:				Sometimes insufficient 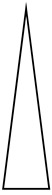 Favorable	Relatively easy 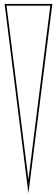 Rarely difficult
Menghini	Expect (Boston Scientific)	19, 22, 25G	2011		
	SonoTip Pro Control (MediGlobe)	19, 22, 25G	2012		
	EUS Sonopsy CY (HAKKO)	21G	2013		
	Expect Slimline (Boston Scientific)	19, 22, 25G	2014		
	EZ shot 3 plus (Olympus)	19, 22, 25G	2016		
EUS-FNB:					
Reverse-bevel	Echo Tip ProCore (Cook Medical)	19, 22, 25G	2012		
Forward-bevel	Echo Tip ProCore (Cook Medical)	20G	2016		
Fork-tip	SharkCore (Medtronic)	19, 22, 25G	2020		
Franseen	Acquire (Boston Scientific)	19, 22, 25G	2016		
	Sono Tip Top Gain (Medi-Globe)	19, 22, 25G	2020		

**Table 4 diagnostics-12-00753-t004:** Immunohistochemical and genetic abnormalities in PanNENs.

			IHC Abnormalities				Genetic Abnormalities				
Author Year	PanNET/ PanNEC	SSTR2A (IHC)	DAXX (IHC)	ATRX (IHC)	Rb (IHC)	p53 (IHC)	*DAXX*	*ATRX*	*KRAS*	*RB1*	*TP53*
Yachida 2012 [82]	PanNET		9.1% (1/11)	36.4% (4/11)	0% (0/11)	0% (0/11)			0% (0/11)	0% (0/11)	0% (0/11)
Marinoni 2014 [83]	PanNET		25% (23/92)	18% (20/92)			*DAXX* or *ATRX* 48% (12/25)				
Gleeson 2017 [84]	PanNET						11.1% (10/90)	10.0% (9/90)	3.3% (3/90)	2.2% (2/90)	3.3% (3/90)
Chan 2018 [85]	PanNET						25% (16/64)	10.4% (7/64)			
Hackeng 2021 [86]	PanNET		DAXX or ATRX 31.1% (208/668)		-						
Simbolo 2021 [87]	PanNET						24.1% (7/29)	20.7% (6/29)			
Hijioka 2017 [88]	PanNET-G3				0% (0/21)				0% (0/21)		
Konukiewitz 2017 [89]	PanNET-G3	77.8% (7/9)	33.3% (3/9)	11.1% (1/9)	0% (0/9)	0% (0/9)					0% (0/9)
Yachida 2012 [82]	PanNEC		0% (0/19)	0% (0/19)	73.7% (14/19)	94.7% (18/19)			28.6% (2/7)	71.4% (5/7)	57.1% (4/7)
Hijioka 2017 [88]	PanNEC				54.5% (24/44)				48.7% (20/41)		
Konukiewitz 2017 [89]	PanNEC	8.3% (1/12)	0% (0/11)	0% (0/11)	41.7% (5/12)	75.0% (9/12)					66.7% (8/12)

IHC, immunohistochemistry; PanNENs, Pancreatic neuroendocrine neoplasms; PanNET, Pancreatic neuroendocrine tumor; PanNEC, Pancreatic neuroendocrine carcinoma; SSTR 2A, Somatostatin receptor 2A.

**Table 5 diagnostics-12-00753-t005:** Precision medicine for PDAC.

Gene Mutation	Frequency	Author Year	Study Design	Patients	Regimen	Results
HRD	HRD 15% *BRCA 1*: 0.9% *BRCA 2*: 3.5% *PALB2*: 0.2% *ATM*: 2.2% *ATM*: 0.2%	Wattenberg 2020 [90]	Retrospective	g*BRCA 1/2* *PALB2* PDAC	Platinum- based regimen	g*BRCA 1/2, PALB2* Control ORR 58% 21% PFS 10.1 mo 6.9 mo OS 24.6 mo 18.8 mo
		Golan 2019 [92]	Phase 3	*gBRCA* PDAC	Olaparib	Olaparib Placebo PFS 7.4 mo 3.8 mo HR 0.53 (95% CI: 0.35–0.82) OS 18.9 mo 18.1 mo HR 0.91 (95% CI: 0.56–1.46)
MSI-H dMMR	1–2%	Marabelle 2019 [93]	Phase 2	MSI-H/dMMR PDAC	Pembrolizumab	ORR 18.2% (4/22) Median PFS: 2.1 mo (95% CI: 1.9–3.4) Median OS: 4.0 mo (95% CI: 2.1–9.8)
		Le 2017 [94]	Prospective	dMMR PDAC	Pembrolizumab	ORR 62% (5/8) DCR 75% (6/8)
*NTRK* gene fusions	less than 1%	Doebele 2020 [97]	Phase 1/2 Pooled analysis of 3 studies	*NTRK* gene fusions Solid tumors	Entrectinib	ORR 57% DCR 74% Median DOR: 10.4 mo Median PFS: 11.2 mo
		Hong 2020 [98]	Phase 1/2 Pooled analysis of 3 studies	*NTRK* gene fusions Solid tumors	Larotrectinib	ORR 79% DCR 91% Median DOR: 35.2 mo Median PFS: 28.3 mo
*KRAS* G12C mutation	unknown	Hong 2020 [100]	Phase 1	*KRAS* G12C mutation Solid tumors	Sotorasib	ORR 32.2% DCR 88.1% Median PFS 6.3 mo
		Skoulidis 2021 [99]	Phase 2	*KRAS* G12C mutation Lung cancer	Sotorasib	ORR 37.1% DCR 80.6% Median DOR 11.1 mo

ORR, objective response rate; DCR, disease control rate; PFS, progression-free survival; OS, overall survival, HR, hazard ratio; DOR, duration of response; mo, months.

**Table 6 diagnostics-12-00753-t006:** Next generation sequencing using EUS-TA specimen.

Author Year	Number of Patients	Puncture Needles	Targeted Panel	Requirements for NGS	DNA Amount/ Concentration Extracted	Success Rate/ Adequacy Rate for NGS	Frequency of Genomic Alternations (PDAC)
Young 2013 [72]	PDAC n = 18 AC NOS n = 2 MCC n = 2 PanNEC n = 1	NA	Custom panel 287 genes	Tumor cells: 20% DNA amount: 50 ng	NA	100% (23/23)	KRAS 83% CDKN2A 44%
Kameta 2016 [73]	PDAC n = 27	NA	Ampliseq Cancer Hotspot Panel v2 50 genes	NA	NA	100% (27/27)	KRAS 96% TP53 44% SMAD4 11% CKDN2A 11%
Gleeson 2016 [74]	PDAC IPMC AC n = 47	NA	Human Comprehensive Cancer GeneRead DNAseq Targeted Panel V2 160 genes	Tumor cells: 20% DNA concentration: 5 ng/μl	Smear cytology: mean 21.0 ng/μL (Range 0–88.7) FFPE: mean 66.9 ng/μL (Range 9.3–164)	61.7% (29/47)	KRAS 93.1% TP53 72.4% SMAD4 31% GNAS 10.3%
Elhanafi 2019 [75]	PDAC n = 167	EUS-FNA/ EUS-FNB 22 G	TruSeq Amplicon Cancer Panel 47 genes	Tumor cells: 10%	NA	70.1% (117/167) * EUS-FNA: 66.9% (97/145) * EUS-FNB: 90.9% (20/22) *	KRAS 88% TP53 68% SMAD4 16%
Park 2020 [76]	PDAC n = 190	EUS-FNA/ EUS-FNB 19,22,25G	Cancer Scan version 1 183 genes	Tumor cells: 30%	NGS success: 1.42 ± 1.57 μg NGS failure: 0.54 ± 1.70 μg	57.4% (109/190)	KRAS 78.9% TP53 60.6% SMAD4 30.3% CKDN2A 25.7%
Ishizawa 2020 [77]	PC n = 26	EUS-FNA/ EUS-FNB 22G	AmpliSeq Comprehensive Cancer Panel 409 genes	NA	mean 171 ng (Range 34–478)	100% (26/26)	KRAS 92% TP53 50% SMAD4 31% CDKN2A 15%
Carrara 2021 [79]	PDAC: 33	EUS-FNB 22G	AmpliSeq Comprehensive Panel v3 161 genes	NA	NA	97.0% (32/33)	KRAS 94% TP53 78% SMAD4 13% CDKN2A 9% GNAS 9%
Habib 2021 [80]	PDAC: 56	NA	Ampliseq Custom Panel 9 genes	DNA concentration: 3.3 ng/μL	NA	100% (56/56)	KRAS 85.7% TP53 32.1% SMAD4 3.6% CKDN2A 3.6%
Larson 2018 [102]	PDAC: 74 ACC: 1 AC: 1	NA	FoundationOne CDx 324 genes	Tumor cells: 20% Specimen surface area: 25 mm^2^	NA	EUS-FNA: 42.9% (3/7) ** EUS-FNB: 70.4% (38/54) **	NA
Kandel 2021 [103]	PDAC: 37 PanNET: 5 Other malignancies: 3 Benign: 5	EUS-FNA: 25G EUS-FNB: 19, 22G	FoundationOne CDx 324 genes	Tumor cells: 20–30% Specimen surface area: 25 mm^2^	EUS-FNA: mean 3.36 ng/μL EUS-FNB: mean 5.93 ng/μl	EUS-FNA:14% ** (7/50) EUS-FNB: 78% (39/50)	NA

PDAC, Pancreatic ductal carcinoma; PanNEC, Pancreatic neuroendocrine carcinoma; IPMC, Intraductal papillary mucinous neoplasm; MCC, Mucinous adenocarcinoma; AC, Ampullary carcinoma; PC, Pancreatic cancer; ACC, Acinar cell carcinoma; PanNET, Pancreatic neuroendocrine tumor; FFPE, Formalin-fixed paraffin-embedded. * Rate of adequate specimen; ** Rate of adequate specimen for FoundationOne CDx (NGS has not been actually performed).

**Table 7 diagnostics-12-00753-t007:** Predictors for adverse events of EUS-tissue acquisition for solid pancreatic lesions.

Predictors	Adverse Events (AEs)
FNA vs. FNB	Comparable
Needle size	Frequency of AEs (Same for both EUS-FNA and EUS-FNB) 25G < 22G < 19G
Number of passes	Possibility of increase in pancreatitis
To-and-fro movement	
Types of SPLs: PanNET	
Pancreatic body or tail cancers	Increase in needle tract seedings

FNA, fine-needle aspiration; FNB, fine-needle biopsy; SPLs, Solid pancreatic lesions; PanNET, pancreatic neuroendocrine tumor.

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
