# Peer review of "The Utility of Endoscopic-Ultrasonography-Guided Tissue Acquisition for Solid Pancreatic Lesions"

_diagnostics, 2022, doi:10.3390/diagnostics12030753_

Round 1

Reviewer 1 Report

The authors provide well and organised review EUS-TA for solid pancreatic lesions. The author should summarize the benefits and faults of these devices and methods to improve the diagnostic performance in Table 1. 

Reviewer 2 Report

This is an excellent updated review on tissue acquisition in solid pancreatic masses. The review is comprehensive. 

Minor comments 

Many centers are using FNB needles in current practice. 

  1. Any difference in diagnostic yield comparing 22g vs. 25g FNB needles
  2. Any comparative studies comparing ROSE and MOSE while using FNB needles

Authors can comment and cite any available references in this regard. 

Reviewer 3 Report

I’ve carefully read the manuscript entitled “The Utility of Endoscopic Ultrasonography-Guided Tissue Acquisition for Solid Pancreatic Lesions” which describes the diagnostic performance and safety of endosonography guided-TA for pancreatic masses. Its strong point is the section on precision medicine.

The review is a good update of the recent literature on the topic, however in order to improve its content for the reader, as there are many such reviews already available, I suggest the following:

  1. Widen the discussion on differential diagnosis, focusing on the change over time regarding solid pancreatic lesions (SPL) – while a few decades ago over 90% of SPLs were considered pancreatic ductal adenocarcinoma, recent data have shown that up to 1 in 4 SPL is non-PDAC lesion. Along with the already discussed differentials, lymphoma, focal pancreatitis and other rare lesions should be also mentioned. A graph/figure on this differential and a timeline with needle designs would be recommended.
  2. In the table on methods to improve diagnostic accuracy of EUS-TA, use of ancillary techniques should be included (contrast-enhancement, elastography). Elastography guidance for TA should be discussed in text also, after the paragraph on contrast enhancement.
  3. Regarding needle passes, the decreased number of passes with FNB needles should be discussed.
  4. The complications should be more thoroughly approached, maybe include a table on this also.
  5. Finally, a summary of recommendations from the authors (maybe in the form of a figure), based on currently available evidence, would provide added value for the paper – regarding needle size, puncture technique and sample processing.

Round 2

Reviewer 3 Report

There is significant improvement with the revised version of the manuscript. The authors addressed all the concerns I pointed out in my first review.